# Impact of a Bundle of Interventions on Quality-of-Care Indicators for *Staphylococcus aureus* Bacteraemia: A Single-Centre, Quasi-Experimental, Before–After Study

**DOI:** 10.3390/antibiotics13070646

**Published:** 2024-07-12

**Authors:** Giacomo Casalini, Cristina Pagani, Andrea Giacomelli, Laura Galimberti, Laura Milazzo, Massimo Coen, Serena Reato, Beatrice Caloni, Stefania Caronni, Simone Pagano, Samuel Lazzarin, Anna Lisa Ridolfo, Sara Giordana Rimoldi, Andrea Gori, Spinello Antinori

**Affiliations:** 1III Division of Infectious Diseases, ASST Fatebenefratelli-Sacco, Luigi Sacco Hospital, 20157 Milan, Italy; giacomo.casalini@asst-fbf-sacco.it (G.C.); serena.reato@unimi.it (S.R.); beatrice.caloni@unimi.it (B.C.); stefania.caronni@unimi.it (S.C.); simone.pagano@unimi.it (S.P.); samuel.lazzarin@unimi.it (S.L.); annalisa.ridolfo@asst-fbf-sacco.it (A.L.R.); spinello.antinori@unimi.it (S.A.); 2Clinical Microbiology, Virology and Bioemergency Diagnostics, ASST Fatebenefratelli-Sacco, Luigi Sacco Hospital, 20157 Milan, Italy; cristina.pagani@asst-fbf-sacco.it (C.P.); laura.galimberti@asst-fbf-sacco.it (L.G.); laura.milazzo@asst-fbf-sacco.it (L.M.); sara.rimoldi@asst-fbf-sacco.it (S.G.R.); 3Department of Biomedical and Clinical Sciences, Università degli Studi di Milano, 20157 Milan, Italy; andrea.gori@unimi.it; 4I Division of Infectious Diseases, ASST Fatebenefratelli-Sacco, Luigi Sacco Hospital, 20157 Milan, Italy; massimo.coen@asst-fbf-sacco.it; 5Centre for Multidisciplinary Research in Health Science (MACH), Università degli Studi di Milano, 20122 Milan, Italy; 6II Division of Infectious Diseases, ASST Fatebenefratelli-Sacco, Luigi Sacco Hospital, 20157 Milan, Italy

**Keywords:** *Staphylococcus aureus* bacteraemia, SAB, antimicrobial stewardship, quality-of-care indicators, infectious disease consultations, quality improvement

## Abstract

*Staphylococcus aureus* bacteraemia (SAB) is a life-threatening bloodstream infection. Improved adherence to quality-of-care indicators (QCIs) can significantly enhance patient outcomes. This quasi-experimental study evaluated the impact of a bundle of interventions on QCI adherence in adult patients with SAB. Additionally, a molecular rapid diagnostic test (mRDT) for *S. aureus* and methicillin resistance was introduced during weekdays. We compared pre-intervention (January–December 2022) and post-intervention (May 2023–April 2024) data on QCI adherence and time to appropriate treatment. A total of 56 and 40 SAB episodes were included in the pre- and post-intervention periods, respectively. Full QCI adherence significantly increased from 28.6% to 67.5% in the post-intervention period (*p* < 0.001). The mRDT diagnosed SAB in eight cases (26.6%), but the time to achieve appropriate target therapy did not improve in the post-intervention period (54 h (IQR 30–74) vs. 72 h (IQR 51–83), *p* = 0.131). The thirty-day mortality rate was comparable between the two periods (17.9% vs. 12.5%, *p* = 0.476). This study demonstrates that a bundle of interventions can substantially improve adherence to SAB management QCIs.

## 1. Introduction

*Staphylococcus aureus* is a major cause of community- and hospital-acquired bacteraemia (SAB), a severe and potentially lethal infection. It often has a complicated course with metastatic foci of infection, such as deep abscesses, endocarditis, or osteomyelitis, and the associated mortality rate can exceed 20–30%, even with the best available therapy [1]. Guidelines define the standard of care for SAB, indicating the diagnostic and therapeutic measures that should be undertaken in all patients, including bedside infectious disease (ID) consultation, follow-up blood cultures, echocardiography, source control, and adequate and timely antibiotic treatment [2,3].

Quality-of-care indicators (QCIs) are measurable elements of clinical practice that can be used to measure the quality of care of a healthcare process, such as SAB management [4]. Assessing the quality of care is crucial for designing and implementing specific antimicrobial stewardship (AMS) interventions to improve healthcare outcomes. SAB, a complex infection that requires various diagnostic and therapeutic measures, serves as an ideal model for creating targeted AMS interventions. The role of the ID consultation is well established in SAB, being associated with a reduction in mortality rates [5]. Several studies have demonstrated the effectiveness of various interventions in improving SAB management, including increased adherence to QCIs and reduced mortality rates. These interventions, often prompted by rapid notification of *S. aureus* growth in blood cultures, have included phone calls with recommendations (by pharmacists or clinical microbiologists) and ID bedside or telemedicine consultations and automatic alerts on the electronic medical record (EMR) providing instruction for SAB management, usually with targeted educational programmes [6,7,8,9,10,11,12,13,14].

In recent years, molecular diagnostics have revolutionised microbiology laboratories, enabling molecular rapid diagnostic tests (mRDTs) to deliver reliable results with a shorter turnaround time than traditional microbiology methods. Some available mRDTs can identify the presence of *S. aureus* and methicillin-resistant *S. aureus* (MRSA) from blood cultures with Gram-positive cocci in clusters, with sensitivity and specificity close to 100% [15]. A 2017 systematic review with meta-analysis demonstrated that using mRDTs in bloodstream infections is associated with a significant reduction in the risk of death, shorter time to the initiation of effective therapy, and reduced hospital length of stay [16]. Furthermore, a more recent systematic review with network meta-analysis showed that the survival benefit in patients with SAB is even greater if mRDTs are implemented in settings where antimicrobial stewardship programmes (ASPs) are already well established [17].

We present the results of a quasi-experimental study to assess the impact of a bundle of interventions on adherence to selected QCIs in adult patients with SAB. The impact of an mRDT on time to appropriate treatment was also evaluated. The aim was to improve the quality of care delivered to patients with SAB, measured as an increase in QCI adherence and time to appropriate treatment after bundle implementation.

## 2. Results

### 2.1. Characteristics of the Study Population

Fifty-six SAB cases were recorded in the pre-intervention period and forty in the post-intervention period. The characteristics of the study population are summarised in Table 1. The patients were predominantly adult males with a significant number of comorbidities, as indicated by the high proportion of patients with a Charlson Comorbidity Index ≥ 2 in both the study periods. The most common comorbidity was cardiovascular disease, followed by chronic pulmonary disease and diabetes mellitus. Few patients were intravenous drug users. Most SAB cases were community-acquired and diagnosed in the emergency department (ED), with the first blood culture drawn upon ED admission. Few cases were diagnosed in the intensive care unit (ICU). The most common source of bacteraemia was venous catheters (19.6% and 22.5% in the pre- and post-intervention periods, respectively), followed by skin and soft tissue infections and pneumonia. Notably, a significant proportion of SAB cases in both study periods had an unknown source (21.4% and 25%, respectively). The proportion of SAB caused by methicillin-resistant *S. aureus* (MRSA) was high throughout the study period (32.1% and 27.5%, respectively). Most SAB cases were complicated, as defined by the Infectious Disease Society of America (IDSA) [2].

### 2.2. Quality-of-Care Indicator Analysis

In the pre-intervention period, adherence to QCIs was notably low, with full compliance achieved in less than 30% of cases (Table 2). ID consultations were conducted in less than half the patients (48.6%). Suboptimal timing of follow-up blood cultures was observed in over half of the cases, leading to inadequate treatment duration in 43% of cases. Although the performance of echocardiography was satisfactory (83.3%), the appropriateness of antibiotic therapy was generally low, especially in MSSA SAB cases (68.4%), where anti-staphylococcal beta-lactams were seldom administered.

A comparison of adherence to selected QCIs was conducted between the pre- and post-intervention periods following bundle implementation. There was a significant increase in full QCI adherence from 28.6% to 67.5% in the post-intervention period (*p* < 0.001) (Table 2). Adherence rates to all QCI measures improved, particularly in ID consultation (48.6% vs. 100%, *p* < 0.001) and follow-up blood cultures (83.6% vs. 100%, *p* = 0.009), with a statistically significant increase. Notably, the timing of ID consultation and follow-up blood cultures was optimal in most cases. Performance in source control did not significantly improve in the post-intervention group (72% vs. 80%, *p* = 0.786), despite being hindered by a high proportion of cases with non-removable focus (33.9% and 25% in the pre- and post-intervention periods, respectively) and unknown focus (19.6% and 25% in the pre- and post-intervention periods, respectively). Echocardiography was performed in the majority of SAB cases in both study periods (83.3% and 95% of cases, *p* = 0.157) and was typically conducted within 7 days of SAB diagnosis. However, transoesophageal echocardiography (TEE) was uncommon, being performed in 22% of cases in the pre-intervention period and 27.5% in the post-intervention period. Antibiotic therapy was appropriate in most cases (76.8% and 90%, *p* = 0.161), although did not significantly improve after the introduction of the bundle. Appropriateness was higher for MRSA bacteraemia than for MSSA in both study periods (all MRSA bacteraemia cases in the post-intervention period received appropriate treatment). Furthermore, in the post-intervention period, early antibiotic treatment switches to oxacillin or cefazolin were performed in most cases (79.3%). When considering the appropriate duration of treatment according to the type of SAB (complicated or uncomplicated), QCI improved after the intervention from 57.1% to 75.7%, although it was not statistically significant.

The mRDT diagnosed SAB in only eight cases (26.6%, including two cases of MRSA) in the post-intervention period. The results of the mRDT and conventional microbiological identification were coincident in all cases. The median time from SAB onset and the first anti-staphylococcal antibiotic treatment improved after bundle introduction, from 46 h (IQR 19–72) in the pre-intervention period to 29 h (IQR 15–54) in the post-intervention period (*p* = 0.168). The median time from SAB onset and definite appropriate antibiotic therapy was 27 h (IQR 8–48) in SAB cases diagnosed with the mRDT first. However, there was no improvement in the time to achieve appropriate target therapy in the post-intervention period (54 h (IQR 30–74) vs. 72 h (IQR 51–83) in the pre-intervention period, *p* = 0.131), even when considering infections by MSSA and MRSA separately (Table 3). The 30-day mortality did not significantly differ in the two study periods (10/56—17.9% vs. 5/40—12.5%, *p* = 0.476).

## 3. Discussion

The implementation of a bundle of interventions for managing SAB, informed by QCIs identified in the literature, improved the quality of care for adult patients hospitalised with SAB. Both overall and individual adherence to QCIs improved, notably in terms of ID consultation and follow-up blood cultures. However, the implementation of the bundle did not significantly affect mortality, and the use of the mRDT did not improve the time to effective antibiotic treatment.

Numerous quasi-experimental studies have investigated the impact of AMS interventions on SAB management, but comparisons between studies are challenging due to variability in both the study population and the interventions evaluated. These interventions, including phone calls with recommendations, structured ASP with written recommendations directly prompted in the EMR, and telemedicine consultations, often lead to improved adherence to QCIs, such as the performance of follow-up blood culture, early source control or appropriate definite treatment, singularly or as a bundle [6,7,8,9,10,11,12,13,14]. The impact on mortality rates shows significant variation across studies, with some reporting statistically significant reductions, but others not observing such a benefit [6,9,10,11,12,13,14]. Despite this extensive evidence, adherence to individual recommendations and the frequency of ID consultation requests in patients with SAB remain suboptimal in clinical practice [18]. Moreover, there is significant variation in practices among different ID specialists, as highlighted by a recent survey of US ID specialists [5,19,20].

One of the paramount QCIs is ID consultation for all patients with SAB, which is associated with an increase in adherence to all other QCIs and ultimately linked to reduced mortality, especially if provided within 48 h [5,21]. The earlier ID consultation is provided, the greater its positive impact [22]. In the post-implementation phase of our study, every SAB case received management support from the ID consultant within 24 h of SAB diagnosis in all cases, except one. This performance was comparable to or even better than previous studies, where ID consultation rates ranged from 80% to 100% [6,13,18]. Our study found that ID consultation showed the most significant improvement after implementing the bundle, suggesting that ID consultation greatly influenced overall adherence to QCIs. However, 40% of SAB cases were admitted to the ID ward, where the effect of ID consultation was not applicable. Finally, we acknowledge that recent studies on ASP implementations are moving beyond bedside consultations for SAB. They are exploring structured ways to deliver consultancy through phone or telemedicine, tools that can be crucial in optimizing the time and resources needed for AMS activities [7,9]

Following bundle implementation, there was a notable increase in the rate of follow-up blood cultures performed (from 83.6% to 100%) and in the timing of blood culture collection, 80% of them being collected at least every 48 h. This enhancement likely contributed to a concurrent improvement in the appropriateness of antibiotic treatment duration, calculated from the first negative follow-up blood culture (increasing from 57.1% to 75.7%). Data from the literature are consistent with our results, demonstrating that the performance of follow-up blood cultures is one of the QCIs that benefits most from bundle implementation [6,7,8,9,10]. The importance of documenting the clearance of blood cultures is demonstrated by a large European cohort of over 900 patients, where persistent positivity of blood cultures for at least one additional day after the initiation of appropriate antibiotic therapy was observed in 32% of cases. Among these patients, the 30-day mortality rate was twice as high compared to those whose blood cultures cleared immediately (28% vs. 13%) [23].

Although the performance of source control increased from 72% to 80% following bundle implementation, it is important to note that the timing of certain types of source control (such as abscess drainage or catheter removal) remained suboptimal even after bundle implementation (within 24 h from SAB diagnosis in slightly more than half the cases). Additionally, a significant proportion of SAB cases (25%) had an unknown source, which obviously could not be addressed through removal. The criticality of timely source control is underscored by an observational study involving over 800 patients with SAB, where each additional day of bacteraemia without source control was associated with a relative risk of death of 1.16 [24]. However, it is worth noting that the rate of SAB cases of unknown origin in our study is consistent with findings reported in the literature, ranging from 10% to 30% [6,8,18]

Echocardiography was a well-established practice among clinicians in our study regarding SAB cases, and its utilization further improved after implementing the bundle. However, transoesophageal echocardiography (TEE) was performed in only 27.5% of cases during the post-intervention period, a limitation of our study that could have led to the missing of some cases of endocarditis. Thus, significant efforts are needed to enhance the rate of TEE, given its substantially higher sensitivity compared to transthoracic echocardiography (TTE) for identifying endocarditis [25].

The appropriateness of antibiotic therapy improved following bundle implementation, considering both antibiotic selection (with anti-staphylococcal beta-lactams being more frequently administered in cases of MSSA SAB and treatment switches being performed earlier upon sensitivity results notification) and duration (Table 2). Adherence to this last QCI increased from 57.1% to 75.7%. This performance is similar to that reported in a large multicentre prospective study (88.6% in uncomplicated SAB and 61.2% in complicated SAB) that evaluated the impact of QCI adherence on SAB prognosis [18]. In pre–post studies focused on ASP implementation, the appropriateness of antibiotic treatment duration was even higher (i.e., 83.6% in the study by Brotherton A, 88% in the study by Veillette J.J., 90% in the study by Berger N.J) [9,10,13]. In our study, treatment duration appropriateness was evaluated based on the definition of complicated or uncomplicated SAB, as proposed by the IDSA in the 2011 guideline. However, some authors argue that this paradigm has a major limitation in that it encompasses several characteristics (host-related, bacteraemia-related, and clinical course-related) that elevate the risk of metastatic infection, but do not definitively indicate the presence of a metastatic focus. Consequently, there is a risk of treating a patient as if they have complicated SAB in the absence of evidence supporting the presence of a secondary site of infection [2]. In a recent article by Ilse J.E. Kouijzer and colleagues, a novel paradigm for managing SAB is introduced. This paradigm emphasizes risk stratification for metastatic infection, the adoption of a personalised diagnostic approach to identify secondary foci, and tailoring treatment based on individual patient characteristics. While promising, this proposal is still in its developmental stages, and further studies are required to assess its efficacy and influence on patient outcomes [26].

The positive impact of implementing mRDTs for bloodstream infections was noted in a 2017 meta-analysis [16]. However, a more recent systematic review and network meta-analysis focused on SAB showed that using mRDTs within an ASP resulted in improved survival, reduced length of stay, and shorter time to optimal treatment (TOT) [16,17]. In our study, the introduction of an mRDT did not lead to a significant improvement in either the time to the first anti-staphylococcal antibiotic or the time to definitive appropriate antibiotic therapy (i.e., TOT). These outcomes may be attributed to the restricted number of SAB cases diagnosed with the mRDT, influenced by the availability of tests due to funding limitations and the timing of mRDT execution. In our study, blood cultures yielding Gram-positive cocci outside of regular working hours (8 a.m. to 4 p.m.) and weekdays (Monday to Friday) were managed using conventional methods, potentially resulting in missed opportunities for prompt SAB diagnosis with the mRDT. Additionally, delays in conveying the results to the attending physician and in modifying the antibiotic therapy according to test results could have had a role. Finally, our result is consistent with the data from the network metanalysis, where TOT did not differ between the use of mRDT alone and standard microbiology + ASP, suggesting that an established ASP is essential for successful mRDT implementation [17]. Additional improvements in mRDT implementation would be necessary to enhance the optimization of antibiotic treatment in bloodstream infections. These would include extending mRDT to all hospital wards, performing the test on all blood cultures with Gram-positive cocci, improving communication with clinicians, and establishing a therapeutic algorithm for cases with a negative mRDT test, which was not standardised during the study period.

The single-centre, pre–post study design inherently limits the generalizability of results due to variations in baseline patient characteristics between groups and the constraints of retrospective data collection, particularly in the initial phase of the study. Additionally, the small sample could have impacted mortality data. The stringent definition of the QCIs used might underestimate the actual performance of healthcare providers in managing SAB. Lastly, the limitations of the definition of complicated SAB may overestimate the true number of complicated cases, potentially affecting the assessment of appropriate antibiotic treatment duration.

## 4. Materials and Methods

### 4.1. Study Design, Setting, and Participants

We conducted an uncontrolled, quasi-experimental, before–after study at Luigi Sacco Hospital in Milan, Italy. During the study period, our hospital did not have an ASP, and before the introduction of the intervention, ID consultations were proactively conducted only in the ICU. Internal guidelines on SAB management were not available.

The study population included adults aged ≥18 years with a diagnosis of SAB. Patients who died within the first 48 h of the initial positive blood culture and those who did not receive active treatment because of palliative care were excluded.

The pre-intervention phase consisted in a retrospective analysis of SAB cases occurred from January to December 2022. A set of QCIs retrieved from the literature was selected to assess the performance of clinicians in the management of SAB: whether ID consultation, follow-up blood cultures, source control, and echocardiography were performed, and whether the antibiotic therapy administered was appropriate in terms of quality and duration. Full adherence to all QCIs simultaneously was assessed, together with singular QCI performance.

The intervention consisted in the development of a bundle of interventions for SAB management based on selected QCIs (those used in the retrospective analysis). These were the first written guidelines on SAB management available in our hospital. The bundle comprised a brief text and a concise summary table containing key recommendations. The cornerstone of the bundle was the mandatory bedside ID consultation (Table 4).

Bundle implementation started in May 2023 within the Internal Medicine Department and ICU. The text and the summary table were distributed to the clinicians, and a dedicated meeting was arranged to explain the recommendations and their rationale. Subsequently, bundle implementation was expanded to include the emergency department and other wards within the hospital starting from September 2023. Positive blood cultures for *S. aureus* were communicated directly to the ID physician in charge of bundle implementation. A bedside consultation was then arranged, ideally within 24 h. Finally, during the implementation phase, the ID consultant service was internally reorganised, with specific wards being assigned to different consultants in order to build a trust relationship between attending physicians and consultants. QCI adherence was assessed prospectively following bundle implementation, and comparisons between the pre- and post-intervention periods were performed. The anticipated timeframe for the intervention was about 12 months: data analysis was closed on 30 April 2024.

Together with the bundle, a molecular rapid diagnostic test (mRDT) that detects *S. aureus* and methicillin resistance on blood cultures with Gram-positive cocci was implemented on weekdays and standard working hours in the Internal Medicine Department, ICU, and Emergency Department. The mRDT used in the study was the Xpert MRSA/SA BC assay (Cepheid, Sunnyvale, CA, USA), a diagnostic test approved by the US Food and Drug Administration (FDA) and the European Community (CE). The test can identify *S. aureus* from positive blood culture fluid containing Gram-positive cocci in clusters. The test automates DNA extraction and real-time polymerase chain reaction (PCR), with a response time of about one hour. It detects the protein A gene (spa) of *S. aureus* and can differentiate MSSA from MRSA by the presence of both the methicillin-resistance gene (mecA) and the staphylococcal cassette chromosome mec (SCCmec) integrated into the attB site in the *S. aureus* chromosome [27]. The microbiology lab directly communicated the result of the test to the ID physician in charge of bundle implementation. Time to definite appropriate antibiotic therapy was compared between the pre- and post-intervention period to assess the impact of the mRDT.

### 4.2. Definitions

The growth of *S. aureus* in blood culture was always considered significant. SAB was defined as community-acquired if the first positive blood culture was collected within 48 h since hospital admission. Complicated SAB was defined according to the 2011 IDSA guidelines. The duration of treatment was calculated from the first negative blood culture. Adequate duration was defined as a minimum of 14 days for uncomplicated SAB, 28 days for complicated SAB, 6 weeks for SAB and endocarditis, and 8 weeks for SAB and osteomyelitis. For each QCI, the denominator was the number of patients with SAB and the numerator was the number of patients with SAB for whom the QCI was feasible. Reference times were established for each QCI, and analyses were only conducted on patients who were alive on day 3 (for source control), day 5 (for follow-up blood cultures), day 7 (for echocardiography), day 14 (for the duration of treatment in patients with uncomplicated SAB), and day 28 (for the duration of treatment in patients with complicated SAB).

The primary object was to evaluate the proportion of patients with full adherence to all applicable QCIs (full adherence) before (pre-intervention phase) and after (post-intervention phase) the introduction of the bundle. Secondary objectives were the evaluation of individual QCI performance, 30-day in-hospital crude mortality, and the median time from SAB onset to definite appropriate antibiotic therapy, comparing data between the pre- and post-intervention study period.

### 4.3. Statistical Analysis

Continuous variables are expressed as medians and interquartile ranges. Categorical variables are expressed as percentages. Continuous variables were compared using Student’s *t* test or the Mann–Whitney U test, as appropriate. Categorical variables (i.e., QCI adherence) were compared using the chi-squared test or Fisher’s exact test, as appropriate. All the statistical tests were two-tailed and were considered significant at *p* less than 0.05.

## 5. Conclusions

A bundle of interventions for SAB management has shown its effectiveness in enhancing overall adherence to QCIs, even in a hospital without an ASP. In our setting, there is still room for improvement, especially in areas such as source control, appropriateness of treatment duration, investigations into SAB sources, and mRDT implementation. Further studies should focus on optimizing the implementation of AMS interventions and maintaining the achieved results in routine clinical practice after the studies conclude.

## Figures and Tables

**Table 1 antibiotics-13-00646-t001:** Characteristics of the study population.

	Pre-Intervention	Post-Intervention
SAB episodes	56	40
Age, years, median (IQR)	74 (61–83)	64 (47–80)
Gender		
Female	21 (37.5)	14 (35)
Male	35 (62.5)	26 (65)
Charlson Comorbidity Index		
≥2	50 (89.3)	28 (70)
<2	6 (10.7)	12 (30)
Comorbidities		
Cardiovascular disease	37 (66.1)	26 (65)
Chronic pulmonary disease	13 (23.2)	9 (22.5)
Obesity	5 (8.9)	6 (15)
Chronic kidney disease	7 (12.5)	5 (12.5)
Diabetes mellitus	15 (26.8)	9 (22.5)
Immunosuppression	6 (10.7)	6 (15)
Solid cancer	10 (17.9)	5 (12.5)
COVID-19	20 (35.7)	1 (3.3)
Hospital admission in the previous 3 months	18 (32.1)	7 (17.5)
Intravenous drug users	3 (5.4)	2 (5)
Setting of SAB diagnosis		
Emergency department	32 (57.1)	22 (55)
Hospital ward	22 (39.3)	17 (42.5)
Intensive care unit	2 (3.6)	1 (2.5)
Epidemiology		
Community-acquired	35 (62.5)	27 (67.5)
Hospital-acquired	21 (37.5)	13 (32.5)
SAB source		
Venous catheter	11 (19.6)	9 (22.5)
Endocarditis	8 (14.3)	3 (7.5)
Osteomyelitis	4 (7.1)	6 (15)
Pneumonia	11 (19.6)	3 (7.5)
Skin and soft tissue infections	7 (12.5)	7 (17.5)
Other	3 (5.4) ^a^	2 (5) ^b^
Unknown	12 (21.4)	10 (25)
*S. aureus* susceptibility profile		
MRSA	18 (32.1)	11 (27.5)
MSSA	38 (67.9)	29 (72.5)
SAB staging *		
Complicated	34 (60.7)	24 (60)
Uncomplicated	22 (39.3)	16 (40)

Data are presented as number (percentage), unless stated otherwise. SAB, *Staphylococcus aureus* bacteraemia; IQR, interquartile range; MRSA, methicillin-resistant *S. aureus*; MSSA, methicillin-sensible *S. aureus*. ^a^ Two cases of complicated urinary tract infection, one case of deep abdominal abscesses. ^b^ Infection of intravascular devices. * Uncomplicated SAB: no endocarditis, no implanted prothesis, negative follow-up blood culture after 2–4 days, absence of fever after 72 h of active treatment, no metastatic foci.

**Table 2 antibiotics-13-00646-t002:** Quality-of-care indicator adherence before and after the introduction of the bundle.

	Pre-Intervention	Post-Intervention	*p*
Full QCI adherence	28.6% (16/56)	67.5% (27/40)	<0.001
ID consultation			
Yes	48.6% (17/35)	100% (24/24)	<0.001
Yes, within 24 h from SAB diagnosis	14.3% (5/35)	95.8% (23/24)	
Not applicable ^a^	37.5% (21/56)	40% (16/40)	
Follow-up blood cultures ^b^			
Yes	83.6% (46/55)	100% (40/40)	0.009
Yes, every 48 h until the first negative	47.3% (26/55)	80% (32/40)	
Source control ^c^			
Performed, when feasible	72% (18/25)	80% (16/20)	0.786
Catheter removal/abscess drainage within 24 h	53.3% (8/15)	53.8% (7/13)	
Non-removable focus	33.9% (19/56)	25% (10/40)	
Unknown focus	19.6% (11/56)	25% (10/40)	
Echocardiography ^d^			
Yes	83.3% (45/54)	95% (38/40)	0.157
Yes, within 7 days from SAB diagnosis	64.8% (35/54)	87.5% (35/40)	
Appropriate target therapy ^e^	76.8% (43/56)	90% (36/40)	0.161
MSSA	68.4% (26/38)	86.2% (25/29)	
MRSA	94.4% (17/18)	100% (11/11)	
Switch to anti-MSSA therapy ^f^	62.2% (23/37)	86.2% (25/29)	0.058
Within 24 h from AST	48.6% (18/37)	79.3% (23/29)	
Appropriate duration of target therapy ^g^	57.1% (28/49)	75.7% (28/37)	0.119

Data are presented as number (percentage), unless stated otherwise. QCIs, quality-of-care indicators; ID, infectious diseases; SAB, *Staphylococcus aureus* bacteraemia; MRSA, methicillin-resistant S. aureus; MSSA, methicillin-sensible *S. aureus*; AST, antimicrobial susceptibility testing. ^a^ Patients admitted to the infectious disease department. ^b^ Patients were excluded from the analysis if death occurred before day 5 from SAB diagnosis. ^c^ Patients were excluded from the analysis of adherence if death occurred before day 3 from SAB diagnosis. ^d^ Patients were excluded from the analysis of adherence if death occurred before day 7 from SAB diagnosis. ^e^ MSSA: oxacillin or cefazolin; MRSA: vancomycin, daptomycin, ceftaroline, linezolid. ^f^ One patient was excluded from the analysis of adherence in the pre-intervention period because of documented penicillin allergy. ^g^ Treatment duration is calculated since the day of the first negative blood culture: uncomplicated SAB 14 days, complicated SAB 28 days, SAB and endocarditis at least 6 weeks, SAB and osteomyelitis at least 8 weeks; patients were excluded from the analysis of adherence if death occurred before day 14 (uncomplicated SAB) or before day 28 (complicated SAB) from SAB diagnosis.

**Table 3 antibiotics-13-00646-t003:** Time to definite appropriate target antibiotic therapy.

	Pre-Intervention	Post-Intervention	*p*
Time to definite appropriate antibiotic therapy, median, hours (IQR)	72 (51–83)	54 (30–74)	0.13
MSSA	77 (70–100)	70 (51–95)	0.09
MRSA	47 (3–73)	21 (8–46)	0.3

IQR, interquartile range. MRSA, methicillin-resistant *S. aureus*; MSSA, methicillin-sensible *S. aureus*.

**Table 4 antibiotics-13-00646-t004:** Summary table of *S. aureus* bacteraemia bundle (translated in English from the original in Italian).

*Staphylococcus aureus* Bacteraemia (SAB) Bundle of Interventions for In-Hospital Management
Mandatory ID Consultation within 24 h from SAB Diagnosis
Start an empirical antibiotic treatment with activity against *S. aureus* within 24 h from the first positive blood culture
2. Switch to beta-lactams antibiotics in MSSA infections within 24 h from susceptibility testing
3. Perform blood cultures every 48 h until the first negative
4. Identify the source of SAB and perform source control (i.e., remove catheters and drain abscesses within 24 h from SAB diagnosis)
5. Echocardiography within 7 days from SAB diagnosis
6. Optimize antibiotic treatment (vancomycin therapeutic drug monitoring)
7. Adequate duration of antibiotic treatment (since the day of the first negative blood culture) Non complicated SAB: 14 days Complicated SAB: 28 days SAB and endocarditis: at least 6 weeks SAB and osteomyelitis: at least 8 weeks

Non complicated SAB: no endocarditis, no implanted prothesis, negative follow-up blood culture after 2–4 days, absence of fever after 72 h of active treatment, no metastatic foci.

## Data Availability

Data will be made available by the corresponding author on request as long as data protection of the patients can be guaranteed.

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
