# Peer review of "Impact of a Bundle of Interventions on Quality-of-Care Indicators for Staphylococcus aureus Bacteraemia: A Single-Centre, Quasi-Experimental, Before–After Study"

_antibiotics, 2024, doi:10.3390/antibiotics13070646_

Round 1

Reviewer 1 Report

Comments and Suggestions for Authors

The manuscript presents the quasi-experimental study that evaluates the impact of a bundle of interventions on adherence to Quality-of-Care Indicators (QCIs) in patients with Staphylococcus aureus bacteraemia (SAB). Given the serious health risks linked to SAB, the study's focus is particularly relevant. The introduction of a molecular rapid diagnostic test (mRDT) and its impact on QCI adherence and patient outcomes are key highlights of the research. However, there are areas where the manuscript could benefit from further clarification, expansion, and revision to strengthen its overall impact.

·       There is an extra "the" in line 29. Please remove one of them.

·       There are a few grammatical, typo mistakes and formatting which need to be addressed.

·       Providing additional context about previous studies on structured antimicrobial stewardship programs (ASP) and their impact on SAB management would be beneficial for readers to understand the background and significance of your study.

·       Please consider providing additional details regarding the components of the intervention bundle and their implementation process. It would also be beneficial to include a clearer description of the anticipated impact of these interventions on clinical outcomes

·       Please expand on the challenges encountered during mRDT implementation and propose strategies for overcoming these barriers in future studies

·       Please consider expanding the discussion section to include more analysis of factors influencing outcomes, such as the timing of interventions and the availability of mRDTs. Providing a more comprehensive comparison with existing literature would help better contextualize the findings. Additionally, discussing in more detail the implications of the study's findings for clinical practice and suggesting directions for future research would strengthen the manuscript.

·       Please clarify and maintain consistent definitions of Quality-of-Care Indicators (QCIs) and complicated Staphylococcus aureus bacteraemia (SAB) throughout the manuscript for improved clarity and understanding

·       Please consider incorporating recent studies to ensure the findings are contextualized within current literature. This addition would enhance the relevance and timeliness of the manuscript's conclusions.

Comments on the Quality of English Language

There are a few grammatical, typo mistakes and formatting which need to be addressed.

Author Response

Reviewer #1

The manuscript presents the quasi-experimental study that evaluates the impact of a bundle of interventions on adherence to Quality-of-Care Indicators (QCIs) in patients with Staphylococcus aureus bacteraemia (SAB). Given the serious health risks linked to SAB, the study's focus is particularly relevant. The introduction of a molecular rapid diagnostic test (mRDT) and its impact on QCI adherence and patient outcomes are key highlights of the research. However, there are areas where the manuscript could benefit from further clarification, expansion, and revision to strengthen its overall impact.

We thank the reviewer for appreciating the topic of this paper, and we agree that the manuscript can be improved.

  • There is an extra "the" in line 29. Please remove one of them.
  • Thank you, this was corrected.
  • There are a few grammatical, typo mistakes and formatting which need to be addressed.
  • We thank the reviewer for identifying these mistakes, which we have corrected to the best of our ability.
  • Providing additional context about previous studies on structured antimicrobial stewardship programs (ASP) and their impact on SAB management would be beneficial for readers to understand the background and significance of your study.
  • The reviewer's suggestion to include a more detailed description of ASP and relevant examples from the literature has been incorporated into the discussion section. We acknowledge, however, the inherent heterogeneity in studies evaluating AMS interventions. Variations in both the study population and the type of intervention implemented make direct qualitative comparisons between our findings and existing literature challenging. To address this, additional references have been included to provide a more comprehensive contextualization of our research.
  • Please consider providing additional details regarding the components of the intervention bundle and their implementation process. It would also be beneficial to include a clearer description of the anticipated impact of these interventions on clinical outcomes. We thank the reviewer for this useful suggestion.
  • Details about the implementation process were included, together with a clearer description of anticipated outcomes (see introduction).
  • Please expand on the challenges encountered during mRDT implementation and propose strategies for overcoming these barriers in future studies.
  • We thank the reviewer for this useful suggestion; the discussion on mRDT was implemented.
  • Please consider expanding the discussion section to include more analysis of factors influencing outcomes, such as the timing of interventions and the availability of mRDTs. Providing a more comprehensive comparison with existing literature would help better contextualize the findings. Additionally, discussing in more detail the implications of the study's findings for clinical practice and suggesting directions for future research would strengthen the manuscript.
  • We thank the reviewer for this useful suggestion. The discussion was implemented according to the suggestions.
  • Please clarify and maintain consistent definitions of Quality-of-Care Indicators (QCIs) and complicated Staphylococcus aureus bacteraemia (SAB) throughout the manuscript for improved clarity and understanding.
  • The text was revised, and consistent definition of QCIs and complicated SAB were reported.
  • Please consider incorporating recent studies to ensure the findings are contextualized within current literature. This addition would enhance the relevance and timeliness of the manuscript's conclusions.
  • Relevant recent studies on SAB and QCIs were included among the references.

Reviewer 2 Report

Comments and Suggestions for Authors

Minor corrections are there (as comments inserted in the attached file) 

Author Response

We thank the reviewer for the suggestions.

In Table 1, separate rows for gender (male and female) were included.

In Table 1, "comunitary" and "nosocomial" were replaced with "community-acquired" and "hospital-acquired," respectively.

Reviewer 3 Report

Comments and Suggestions for Authors

Addressing hospital-acquired infection is a current and important topic. To improve the article, we suggest the following:

1.      The introduction is too brief and needs to be enhanced.

2.      The results should be presented consistently, with appropriate statistical analysis (including p-values) for each characteristic compared between the two groups; the p-value should be given correctly (p<0.001, p=0.xyz (3 digits)); 1-2 figures can be included.

3.      Materials and Methods: Figure 1 is a table; if it has been cited from a literary source, this citation must be highlighted.

4.      The detected plagiarism percentage exceeds 20%, it needs to be reduced below this value.

5.      The bibliography needs to be improved.

Author Response

Addressing hospital-acquired infection is a current and important topic. To improve the article, we suggest the following:

  1. The introduction is too brief and needs to be enhanced.
  2. We agree with the reviewer and introduction was expanded.
  3. The results should be presented consistently, with appropriate statistical analysis (including p-values) for each characteristic compared between the two groups; the p-value should be given correctly (p<0.001, p=0.xyz (3 digits)); 1-2 figures can be included.
  4. We thank the reviewer for the suggestion. P-values were reported as suggested.
  5. Materials and Methods: Figure 1 is a table; if it has been cited from a literary source, this citation must be highlighted.
  6. “Figure 1”was replaced with “Table 4” as suggested. The table has not been cited by the literature.
  7. The detected plagiarism percentage exceeds 20%, it needs to be reduced below this value.
  8. We thank the reviewer for this indication. We are revised all the text in the attempt to reduce the plagiarism percentage.
  9. The bibliography needs to be improved.
  10. We agree with the reviewer and bibliography was significantly improved with recent literature.

Round 2

Reviewer 1 Report

Comments and Suggestions for Authors

The manuscript presents a quasi-experimental study that evaluates the impact of a bundle of interventions on adherence to Quality-of-Care Indicators (QCIs) in patients with Staphylococcus aureus bacteraemia (SAB). The authors have effectively highlighted the implications of the study findings, emphasizing the novelty of the research and its potential impact on future studies and clinical practice. Also, the references have been updated to include more recent studies, ensuring the context is aligned with current literature. The manuscript has been appropriately revised based on previous suggestions, with additional details incorporated as requested in the earlier version. However, some minor edits/corrections need to be addressed before the next step.

·       Please check the spelling of "Staphylococcus aureus" throughout the manuscript, especially in the title. Additionally, in line 132, the name of the species is spelled incorrectly.

·       It is important to note that if you use American English for certain terms, please use it consistently throughout the entire manuscript. Avoid the usage of both American and British English in a single manuscript. Please check Table 4 for the spelling of "SAB”.

·       Please include the expansion of ASP or explain that ASP stands for Antimicrobial Stewardship Program. Additionally, mention that another commonly used abbreviation is AMS, which you have already used in the manuscript. Before using any abbreviation in the manuscript, please explain or expand the term first to ensure clarity and make it easily understandable to readers.

Author Response

The manuscript presents a quasi-experimental study that evaluates the impact of a bundle of interventions on adherence to Quality-of-Care Indicators (QCIs) in patients with Staphylococcus aureus bacteraemia (SAB). The authors have effectively highlighted the implications of the study findings, emphasizing the novelty of the research and its potential impact on future studies and clinical practice. Also, the references have been updated to include more recent studies, ensuring the context is aligned with current literature. The manuscript has been appropriately revised based on previous suggestions, with additional details incorporated as requested in the earlier version. However, some minor edits/corrections need to be addressed before the next step.

  • Please check the spelling of "Staphylococcus aureus" throughout the manuscript, especially in the title. Additionally, in line 132, the name of the species is spelled incorrectly.
  • We thank the reviewer for pointing out these mistakes, which we have now corrected.
  • It is important to note that if you use American English for certain terms, please use it consistently throughout the entire manuscript. Avoid the usage of both American and British English in a single manuscript. Please check Table 4 for the spelling of "SAB”.
  • We thank the reviewer for pointing out these inconsistencies, which we have now corrected. Spelling of SAB in Table 4 was corrected.
  • Please include the expansion of ASP or explain that ASP stands for Antimicrobial Stewardship Program. Additionally, mention that another commonly used abbreviation is AMS, which you have already used in the manuscript. Before using any abbreviation in the manuscript, please explain or expand the term first to ensure clarity and make it easily understandable to readers.
  • We thank the reviewer for highlighting these issues, which we have now corrected; ASP was expanded in line 67 and AMS in line 46.

Reviewer 3 Report

Comments and Suggestions for Authors

The article has been significantly improved. For improvement, a suggestion. The p-value is usually edited to three decimal places.

Author Response

The article has been significantly improved. For improvement, a suggestion. The p-value is usually edited to three decimal places.

We thank the reviewer for the comment. P-values were updated as suggested.